# miR-129-5p Participates in Hair Follicle Growth by Targeting HOXC13 in Rabbit

**DOI:** 10.3390/genes13040679

**Published:** 2022-04-12

**Authors:** Fan Yao, Bohao Zhao, Shuaishuai Hu, Shaocheng Bai, Rongshuai Jin, Chen Zhang, Yang Chen, Xinsheng Wu

**Affiliations:** College of Animal Science and Technology, Yangzhou University, Yangzhou 225009, China; fanyao1214@163.com (F.Y.); bhzhao@yzu.edu.cn (B.Z.); 18852726848@163.com (S.H.); bsc305972572@163.com (S.B.); rsjinnn@163.com (R.J.); 13733343697@163.com (C.Z.); yangc@yzu.edu.cn (Y.C.)

**Keywords:** miR-129-5p, *HOXC13*, DPCs, proliferation and apoptosis

## Abstract

Mammalian hair formation is critically determined by the growth of hair follicles (HF). MiRNAs are crucial in the periodic development of hair follicles; they maintain epidermal homeostasis by targeting genes and influencing the activity of signaling pathways and related regulators. Our study discovered miR-129-5p to be overexpressed in the skin of Angora rabbits during catagen, and was negatively correlated with *HOXC13* expression (Pearson’s R *=* −0.313, *p* < 0.05). The dual-Luciferase reporter gene detection system and Western blotting confirmed that miR-129-5p targeted *HOXC13*. In addition, miR-129-5p overexpression was found to significantly inhibit the expression of hair follicle development-related genes (HFDRGs), such as *BCL2*, *WNT2*, *CCND1,* and *LEF1* (*p* < 0.01), and promoted the expression of *SFRP2*, *TGF-β1,* and *FGF2* (*p* < 0.01), which was the same as the knockdown of *HOXC13*. In contrast, the knockout of miR-129-5p was the opposite, and it demonstrated similar results to the overexpression of *HOXC13*. CCK8 and flow cytometry demonstrated that miR-129-5p mimics significantly promoted the apoptosis of dermal papilla cells (DPCs) and inhibited proliferation (*p* < 0.01), while the inhibitor was found to reduce the apoptosis of DPCs and promote proliferation (*p* < 0.01). These results showed that miR-129-5p can participate in the periodic development of HF by targeting *HOXC13,* and it can induce apoptosis and inhibit proliferation of DPCs. These results will help to understand the role and mechanism of miR-129-5p in the periodic development of HF, and will provide support for subsequent studies, not only providing a theoretical basis for genetically improving the quality of hair in animals in the future, but also a new theory and method for diagnosing and treating hair loss in humans.

## 1. Introduction

MicroRNAs are 20–25 bp long endogenous non-coding small RNAs, which execute mRNA degradation or transcriptional inhibition by binding with the complementary region of the mRNA of the target gene, thus affecting the growth and development of animals [1,2]. Recent studies have identified miRNA as an effective regulator of various biological processes related to hair growth, the hair follicle (HF) cycle, and skin keratinocytes differentiation in animals [3,4]. For example, miR-31 expression increased significantly during the anagen of mouse HF, but the effect was reversed in catagen; the expression of miR-205, which is involved in regulating the periodic growth of HF, was low in anagen of mouse HF, but was the highest in catagen [2,5]. In mouse skin, the overexpression of miR-22 could inhibit HF transformation from telogen to anagen, as well as promote the transformation of HF from anagen to catagen, which finally leads to hair loss in mice [6]. In addition, miR-129-5p is also a regulator of mammalian melanin biosynthesis, which influenced the structure of goat hair color by decreasing the expression of *TYR* [7]. Although miRNA is immediately correlated with the development of HF, the effect of miR-129-5p on HF development remains elusive.

The synergistic relationship between a large number of cells, such as follicle stem cells (FSCs), dermal papilla cells (DPCs), and keratinocytes, determines the growth of hair and HF [8,9]. In addition, there are other signaling pathways that participate in HF growth, such as the Wnt/β-catenin, TGF-β, and Shh signaling pathways [10,11,12]. The *HOX* protein is evolutionarily a highly conserved homologous protein, which is indispensable for developing and regenerating organs [13]. *HOXC13* is majorly involved in morphogenesis, periodic growth, and the formation of hair traits of skin HF. The deletional mutation of the *HOXC13* gene led to PHNED (pure hair and nail ectodermal dysplasia) [14,15]. *HOXC13* overexpression in goat fibroblasts promoted the expression of HFDRGs (hair follicle development-related genes) (*TGF-βR II*, *Rorα2*, *Nanog*, *Wnt10b,* and *PdgfrA*), which positively influenced HF growth, and inhibited the expression of HFDRGs (*Bmp2*, *Msx2*, *Ntrk3*, and *Detal*), which negatively affected HF growth [16]. *HOXC13* blocked the conversion of somatostatin by inhibiting the signal transduction of *TGF-β1* [17,18]. Concurrently, *HOXC13* inhibited the development of mouse HF by targeting *Soat1* [19]. In addition, *HOXC13* directly regulated keratin and keratin-associated proteins to regulate the growth and development of HF. For instance, *HOXC13* controlled the growth and development of cashmere by regulating *KAP9.2* and *KAP11.1* [20]. Additionally, *HOXC13* adjusted the promoter activities of *KRT1*, *KRT2*, *KRT38*, and *KRT84* [21]. Studies have already shown that *HOXC13* exerts its regulatory effect as a transcription factor on HFDRGs, but its upstream regulatory factors are yet to be reported.

MiRNAs are essential for the periodic development of HF. This study has defined the expression patterns and rules of miR-129-5p and *HOXC13*, verified that they targeted *HOXC13*, and highlighted their effects on the proliferation and apoptosis of HFDRGs and DPCs. Our study suggests that miR-129-5p may regulate HF growth by targeting *HOXC13*, and may regulate the proliferation and apoptosis of DPCs.

## 2. Materials and Methods

### 2.1. Animals

This study selected 15 healthy rabbits of the same weight, and their back skin was shaved with a push. During sampling, the rabbits were administered with a Zoletil-50 (6 mg/kg) injection into the ear vein, ensuring that the muscle relaxants used in anesthesia did not inhibit respiration. The tissue samples of the rabbits’ back skin on days 30, 60, 90, 120, 130, and 150 were collected and directly frozen in liquid nitrogen, followed by storage at −80 °C. The rabbits were killed after the last sampling. The animal experiments in this paper complied with animal ethics and were approved by the Animal Care and Use Committee of Yangzhou University.

### 2.2. Cell Culture and Transfection Assay

The rabbit DPCs of hair follicle were isolated as part of a previous study [22], while the RAB-9 cells were bought from ATCC. DPCs were used for subsequent experiments, and the RAB-9 cells were used for the dual-Luciferase report assay only. In an incubator at 37 °C and with 5% CO_2_, the DPCs were cultured in the MSCM complete medium, and RAB-9 cells were cultured in MEM (with 10% FBS). Then, the adherent cells were separated using 0.25% trypsin. When cells grew to an 80% confluence in 24-well plates, pcDNA3.1-HOXC13, siRNA-HOXC13, miR-129-5p mimics, the inhibitor, and the negative control were transfected into the DPCs using Lipofectamine 2000 (Invitrogen, Carlsbad, CA, USA). According to the previous pre-experiment, the final concentration of miR-129-5p was 50 nM, siRNA-HOXC13 was 20 nM, and pcDNA3.1-HOXC13 was 2 ng/mL.

### 2.3. Quantitative Real-Time Polymerase Chain Reaction

According to the manufacturer’s instructions, an RNAsimple Total RNA Kit (Tiangen, China) was used to extract the total RNA from the skin tissues of the Angora rabbits and the DPCs. A HiScript III RT SuperMix for qPCR and miRNA 1st Strand cDNA Synthesis Kit (by stem-loop) (Vazyme, Nanjing, China) was reverse transcribed into cDNA, and the stem-loop primer sequence of miR-129-5p was 5′-GTCGTATCCAGTGCAGGGTCCGAGGTATTCGCACTGGATACGACGCAAGC-3′. A ChamQ SYBR qPCR Master Mix (Vazyme, Nanjing, China) was used for qPCR and a miRNA Universal SYBR qPCR Master Mix (Vazyme, Nanjing, China) was used for qPCR of miRNA. The qPCR primer sequence of miR-129-5p was as follows: F: 5′-CGCTTTTTGCGGTCTGG-3′, R: 5′-AGTGCAGGGTCCGAGGTATT-3′. The data were analyzed using QuantStudio^®^ 5 (Thermo Fisher Scientific, Cleveland, OH, USA) software. The relative levels of gene expression were calculated using the 2^−ΔΔCt^ method [23], while GAPDH was used as a control to normalize the expression of related genes, and U6 was used to normalize the miR-129-5p expression. The primer sequences used are represented in Table 1.

### 2.4. Construction of pcDNA3.1-HOXC13

The rabbit HOXC13 sequence was obtained from NCBI (Gene ID: 100352321). The CDS region of *HOXC13* was cloned from the Angora rabbits’ skin tissues and inserted between *Hind*III and *EcoR*I restriction sites of pcDNA3.1 (F: 5′-ctagcgtttaaacttaagcttATGACGACTTCGCTGCTCCTG-3′, R: 5′-tgctggatatctgcagaattcTCAGGTGGAGTGGAGGTGCG-3′).

### 2.5. Western Blot

A cell lysis buffer for Western and IP (Beyotime, Shanghai, China) was used for collecting the total protein. The protein loading amount was 30 μg, and the protein was detected using ChemiDoc™ Touch (Bio-Rad, Hercules, CA, USA). The following antibodies were used: 1:20000 anti-GAPDH mouse monoclonal antibody (Abcam, Cambridge, MA, USA), and 1:1000 anti-HOXC13 mouse monoclonal antibody (Santa Cruz Biotechnology, Santa Cruz, CA, USA).

### 2.6. Dual-Luciferase Reporter Gene Assay

To construct the HOXC13-3′UTR wild-type plasmid, the *HOXC13* 3′UTR region containing the miR-129-5p binding site was cloned and inserted between the *Spe*I and *Hind*III restriction sites of the pMIR-report Luciferase vector (F: 5′-tgatgaaagctgcgcactagtCTCCACCTCCCTACATCCAGG-3′, R: 5′-aaaagatcctttattaagcttCACTCTGCTACTACTACCTCCCCAC-3′). On the other hand, the binding site on the wild-type plasmid was mutated from CGTTTTT to ATGGCGG using a rapid mutation kit (Vazyme, Nanjing, China), to construct the HOXC13-3′UTR mutant plasmid. The constructed pMIR-report plasmid, miR-129-5p mimics, and pRL-TK were co-transfected into the RAB-9 cells, and 48 h after transfection, 1× PLB (passive lysis buffer) was used to lyse the cells and the double luciferase activity was measured. The relative fluorescence activity of the fluorescent vector plasmid was tested using the Dual-Luciferase Reporter System (Promega, Madison, WI, USA).

### 2.7. Apoptosis Assay

The cells were treated with an Annexin V-FITC Apoptosis Detection Kit (Vazyme, Nanjing, China) and detected using flow cytometry FACSAria SORP (Becton Dickinson, San Jose, CA, USA). The test results were then analyzed using FlowJo V10 (Flowjo, LCC, Ashland, OR, USA) software.

### 2.8. Detection of Cell Proliferation

The proliferation of DPCs was detected by the CCK-8 method, using Cell Counting Kit-8 (Vazyme, Nanjing, China). After 6 h of transfection, the DPCs were digested and inoculated into 96-well plates and cells were recorded as 0 h after adhesion. The absorbance (OD) values of cells at a wavelength of 450 nm were measured at 0, 24, 48, and 72 h using the Infinite M200 Pro (Tecan, Mannedorf, Switzerland), and the cell proliferation curve was subsequently plotted.

### 2.9. Statistical Analysis

The data were analyzed using SPSS 25.0 (SPSS, Inc., Chicago, IL, USA) and plotted using the Graphpad 6.0 (GraphPad Software, San Diego, CA, USA) software. *p* < 0.05 indicated that the data have significant differences, while *p* < 0.01 indicated that the data have extremely significant differences. There were at least three biological and three operational repetitions per experiment, and the error bars represented the mean ± SD.

## 3. Results

### 3.1. MiR-129-5p Regulates HOXC13 Expression

In the previous study, we established the HF synchronization model, then the qRT-PCR results showed that miR-129-5p reached the highest expression level on the 120th day, while *HOXC13* reached the highest expression on the 60th day, which means that miR-129-5p expressed in telogen and *HOXC13* expressed in anagen. The analysis of correlation indicated that the expressions of miR-129-5p and *HOXC13* were significantly negatively correlated (Pearson’s R = −0.313, *p* < 0.05) (Figure 1a). After miR-129-5p was transfected into the DPCs, the expression of the HOXC13 protein in the DPCs was observed by a fluorescence inverted microscope (Figure 1b). The overexpression of miR-129-5p inhibited the expression of the HOXC13 protein, while the knockout of it promoted the expression of the HOXC13 protein in the DPCs.

### 3.2. MiR-129-5p Directly Targets the Regulation of HOXC13

To further investigate the relationship between miR-129-5p and *HOXC13*, according to the prediction of bioinformatics on HOXC13-3′UTR and miRNA targeting binding sites, the binding sites of miR-129-5p and *HOXC13* were determined (Figure 2a). In addition, the Dual-Luciferase Reporter Gene System was constructed to verify that it targeted *HOXC13* 3′-UTR. The luciferase activity of the RAB-9 cells was detected after pMIR-HOXC13-3′-UTR-WT and pMIR-HOXC13-3′-UTR-MUT were co-transfected with miR-129-5p mimics. The luciferase analysis indicated that the overexpression of miR-129-5p significantly decreased the luciferase activity of *HOXC13,* including a wild-type 3′UTR, and the knockdown of it was the opposite (*p* < 0.05) (Figure 2b,c). In addition, miR-129-5p significantly inhibited *HOXC13* mRNA expression (*p* < 0.01) (Figure 2d). However, after the miR-129-5p inhibitor was transfected into the DPCs, the *HOXC13* mRNA expression was found to be markedly up-regulated (*p* < 0.01) (Figure 2e). The Western blot results confirmed that it could inhibit HOXC13 protein expression (Figure 2f).

### 3.3. MiR-129-5p Targeted HOXC13 to Regulate the Expression of the HFDRGs

The DPCs were transfected with pcDNA3.1-HOXC13 and siRNA-HOXC13 to detect the expression of *HOXC13* and the HFDRGs. The results indicated that *HOXC13* could significantly promote the expression of HFDRGs, such as *BCL2*, *WNT2*, *CCND1,* and *LEF1* (*p* < 0.01), and markedly reduce the expression of *SFRP2*, *TGF-β1,* and *FGF2* (*p* < 0.01) (Figure 3a). Conversely, knocking down *HOXC13* was found to markedly reduce the expression of *BCL2*, *WNT2*, *CCND1,* and *LEF1*, as well as inhibit the expression of *SFRP2*, *TGF-β1,* and *FGF2* (*p* < 0.01) (Figure 3b). The DPCs were transfected with the miR-129-5p mimics and inhibitor to detect the expression of miR-129-5p and the HFDRGs. The results indicated that miR-129-5p could markedly reduce the expression of *BCL2*, *WNT2*, *CCND1,* and *LEF1 (p* < 0.01), and markedly promote the expression of *SFRP2* and *TGF-β1* (*p* < 0.01). It was also found to promote the expression of *FGF2* (*p* < 0.05) (Figure 3c). Conversely, knocking down miR-129-5p was found to markedly promote the expression of *BCL2*, *WNT2*, *CCND1,* and *LEF1*, as well as inhibit the expression of *SFRP2*, *TGF-β1,* and *FGF2* (*p* < 0.01) (Figure 3d).

### 3.4. MiR-129-5p Enhances Apoptosis and Inhibits Proliferation of DPCs

The miR-129-5p mimics and inhibitor were transfected into the DPCs to detect the apoptosis and proliferation of the DPCs. The results indicated that the overexpression of miR-129-5p promoted the apoptosis of DPCs (*p* < 0.01) (Figure 4a), while the knockdown of it was the opposite (*p* < 0.01) (Figure 4b). In addition, the overexpression of miR-129-5p inhibited the proliferation of the DPCs from 48 to 72 h (*p* < 0.01) (Figure 4c), while the knockdown of it promoted the proliferation of the DPCs from 24 to 72 h (*p* < 0.01) (Figure 4d).

### 3.5. HOXC13 Inhibits Apoptosis and Enhances the Proliferation of DPCs

PcDNA3.1-HOXC13 and siRNA-HOXC13 were transfected into the DPCs to detect the apoptosis and proliferation of the DPCs. The results indicated that the overexpression of *HOXC13* inhibited the apoptosis of the DPCs (*p* < 0.01) (Figure 5a), while the knockdown of *HOXC13* was the opposite (*p* < 0.01) (Figure 5b). Additionally, the overexpression of *HOXC13* promoted the proliferation of the DPCs from 48 to 72 h (*p* < 0.01) (Figure 5c), while the knockdown of *HOXC13* inhibited the proliferation of the DPCs from 48 to 72 h (*p* < 0.01) (Figure 5d).

## 4. Discussion

Skin is an important organ of mammals, with functions that involve resisting environmental hazards, maintaining homeostasis, preventing cold and warming, camouflage, and the exchange of interspecies information. Animal hair is a by-product of the HF cycle. Both the HF cycles and skin cycles are complexly coordinated by multiple genes and related pathways, along with a large number of regulatory factors. MiRNAs, as endogenous non-coding RNAs, widely exist in animals and regulate the expression of the genes and proteins [25]. Studies have shown miRNAs to significantly affect the morphogenesis, proliferation, and apoptosis of HF, as well as present in various diseases of HF [26,27,28,29,30].

Recent studies have represented that miR-129-5p can significantly affect oncogenes and adjust cancer cells. For example, it had been found that the growth and migration of chondrosarcoma cells could be restrained by targeting *SOX4* and inhibiting glioma cells’ proliferation by targeting *TGIF2* [31,32]. *HOXC13*, which is one of the miR-129-5p target genes and functions as a transcription factor, was primarily related to HF development and occurrence [33]. In addition, *HOXC13* directly regulated keratin and keratin-associated proteins for controlling the growth and development of HF [34,35,36]. *HOXC13* promoted the growth of cancer cells and inhibited their apoptosis [37,38]. In our study, miR-129-5p reached the highest expression level on the 120th day, indicating that it may play a role in catagen [39], while *HOXC13* was found to be at the highest expression level on the 60th day, indicating that it may play a key role in anagen [39], and the expression of miR-129-5p and *HOXC13* is significantly negatively correlated. Our data also showed that miR-129-5p can directly target rabbit HOXC13-3′-UTR to reduce the expression of *HOXC13* and the HOXC13 protein.

In addition, the development of HF was regulated by related genes, such as *WNT2*, *SFRP2*, *TGF-β1*, and *LEF1*, which regulated skin morphogenesis and hair growth significantly [40,41]. Among these genes, *LEF1* possessed a major role in HF morphogenesis and periodic growth [42]. *TGF-β1* regulated the degeneration of HF through high expression [43]. *SFRP2* was a key factor in HF development in catagen [44]. Our study indicated that the expression of these related genes changed after changing the expression of miR-129-5p and *HOXC13*, suggesting that miR-129-5p may affect the expression of these genes to participate in HF development by targeting *HOXC13*.

DPCs provide sufficient nutrition and signal support for the periodic regeneration of HF. This study showed that miR-129-5p restrained apoptosis and reduced the proliferation of DPCs, while the inhibition of it reduced apoptosis and promoted the proliferation of DPCs. However, the results of the overexpression of *HOXC13* were opposite to those of the overexpression of miR-129-5p, indicating that miR-129-5p may regulate the proliferation and apoptosis of DPCs by targeting *HOXC13.*

In summary, miR-129-5p can induce apoptosis and inhibit the proliferation of DPCs, and participate in maintaining skin development, the HF cycle, and hair growth balance by targeting *HOXC13.* These results help to further understand the role and mechanism of miR-129-5p in the periodic development of hair follicles, which not only provides a theoretical basis for genetically improving the quality of hair in animals in the future, but also a new theory and method for diagnosing and treating hair loss in humans.

## Figures and Tables

**Figure 1 genes-13-00679-f001:**
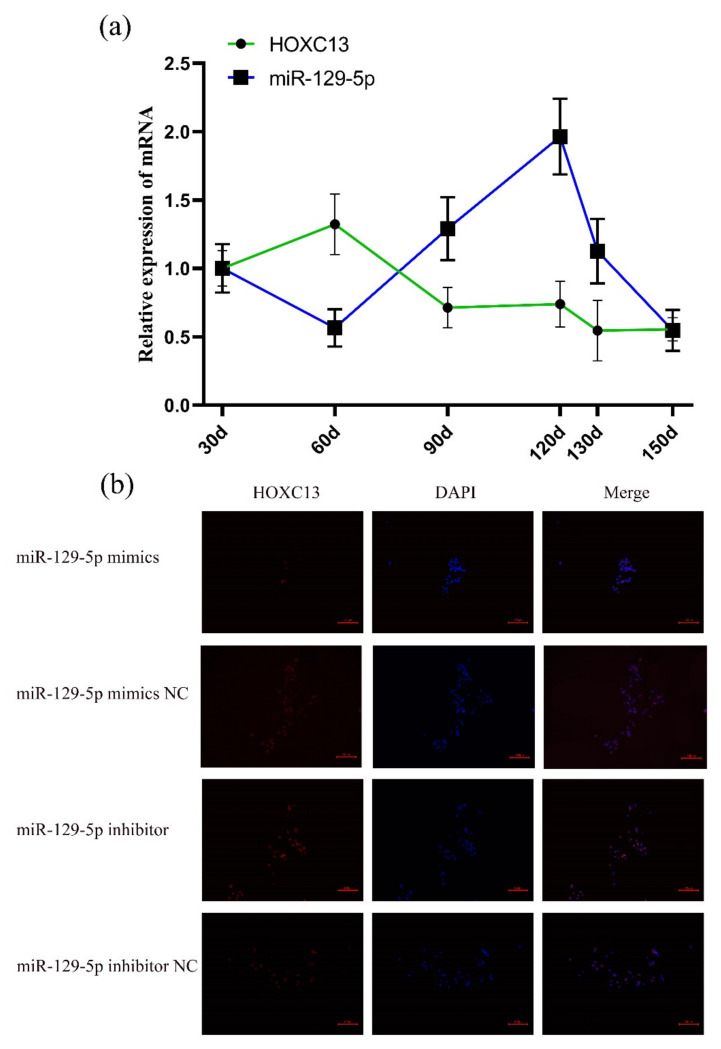
MiR-129-5p regulated *HOXC13* expression. (**a**) The expression of miR-129-5p and *HOXC13* in different stages of HF development. (**b**) MiR-129-5p regulated *HOXC13* expression in DPCs; the scale in Figure 1b is 100 μm.

**Figure 2 genes-13-00679-f002:**
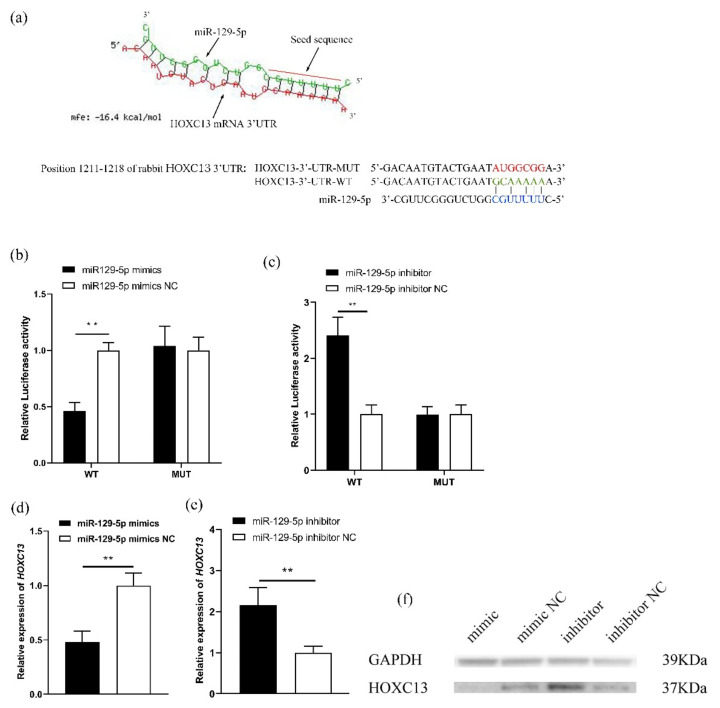
MiR-129-5p directly targeted *HOXC13*. (**a**) Prediction and consequential pairing of *HOXC13* and miR-29-5p, and pMIR-HOXC13-3′-UTR-WT and pMIR-HOXC13-3′-UTR-MUT were constructed according to the assumed miR-129-5p binding site. Forecast of targeted binding sites between miR-129-5p and *HOXC13* used RNAhybrid online software (https://bibiserv.cebitec.uni-bielefeld.de/accessed on 11 April 2022) [24]. (**b**,**c**) The Dual-Luciferase Reporter Gene System verified that miR-129-5p targeted HOXC13 3′-UTR. (**d**,**e**) MiR-129-5p affected the expression of *HOXC13* in DPCs. (**f**) MiR-129-5p affected the expression of the HOXC13 protein in DPCs. ** *p* < 0.01; mfe: minimum free energy; WT: wild-type; MUT: mutant-type; mimic: miR-129-5p mimics; inhibitor: miR-129-5p inhibitor; DPCs: dermal papilla cells.

**Figure 3 genes-13-00679-f003:**
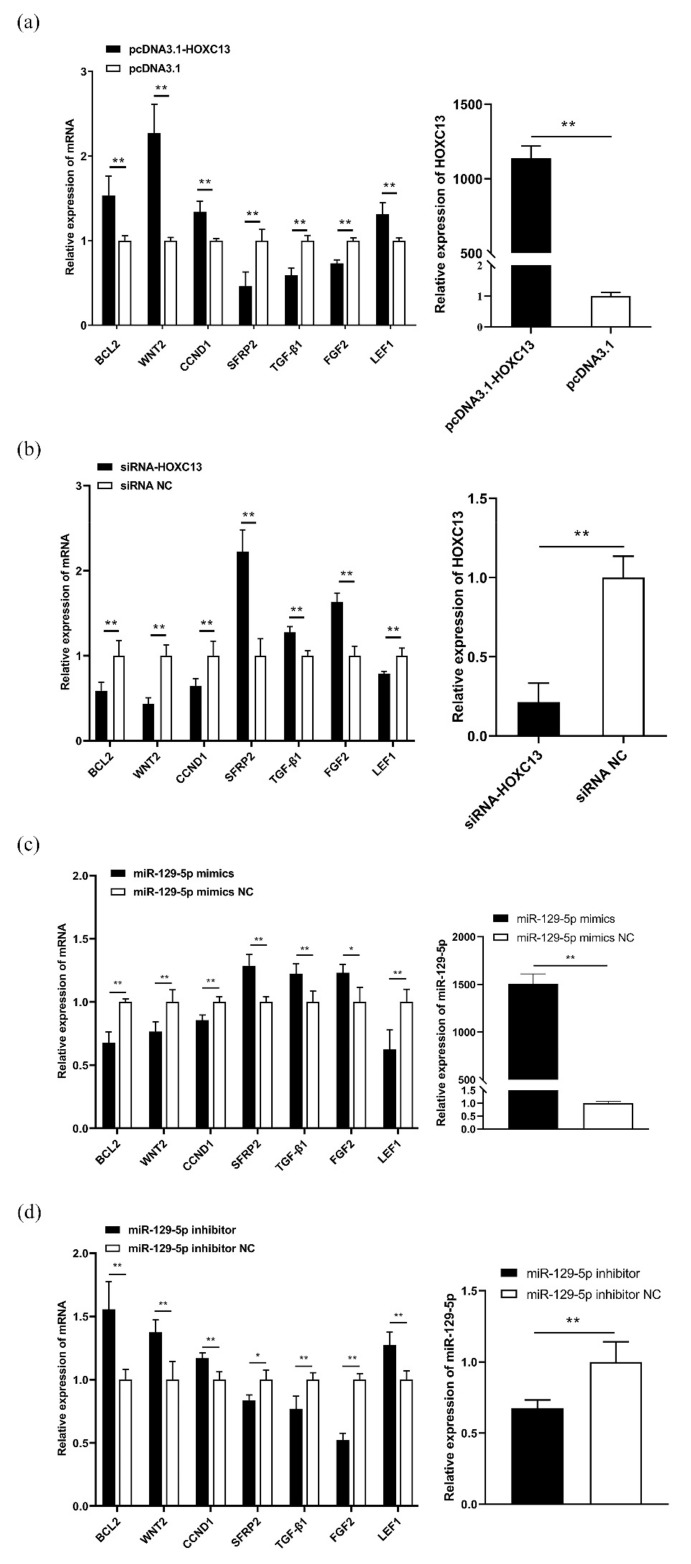
MiR-129-5p and *HOXC13* affected the expression of HFDRGs. (**a**) PcDNA3.1-HOXC13 regulates the mRNA expression of the HFDRGs and overexpression of *HOXC13* in DPCs. (**b**) SiRNA-HOXC13 regulates the mRNA expression of the HFDRGs and the knockdown of *HOXC13* in DPCs. (**c**) MiR-129-5p mimics regulate the mRNA expression of the HFDRGs and overexpression of miR-129-5p in DPCs. (**d**) MiR-129-5p inhibitor regulates the mRNA expression of the HFDRGs and knockdown of miR-129-5p in DPCs. HFDRGs: hair follicle development-related genes. * *p* < 0.05; ** *p* < 0.01.

**Figure 4 genes-13-00679-f004:**
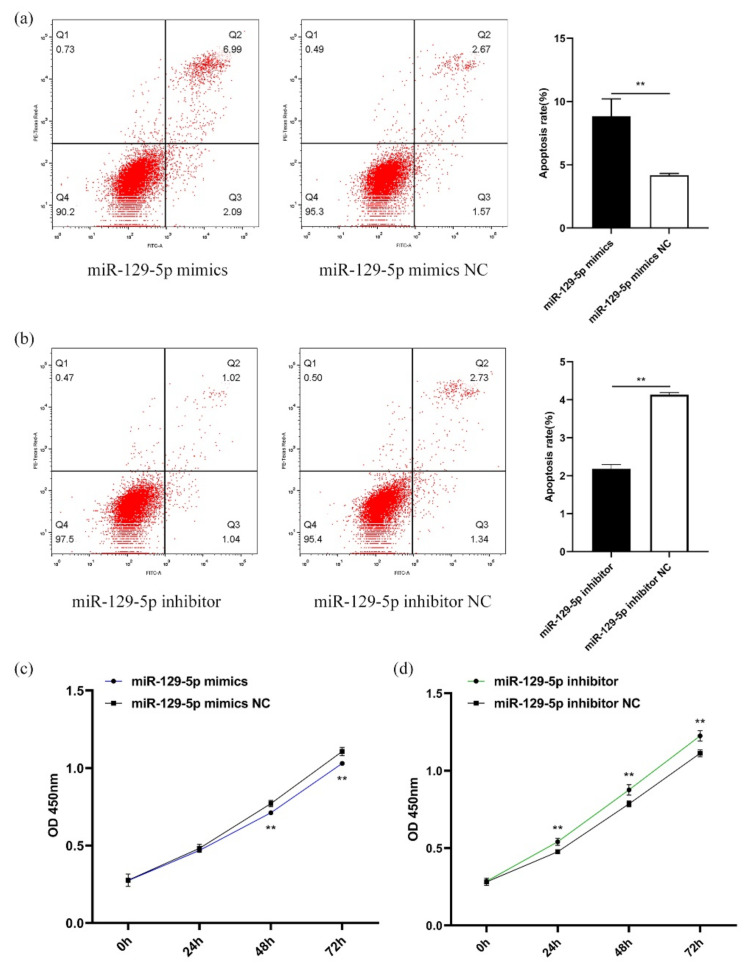
MiR-129-5p promoted apoptosis and inhibited the proliferation of DPCs. (**a**) MiR-129-5p mimics promoted the apoptosis of DPCs. (**b**) MiR-129-5p inhibitor inhibited the apoptosis of DPCs. (**c**) After miR-129-5p mimics were transfected into DPCs, the effect of this on the proliferation of DPCs was determined after 0, 24, 48, and 72 h. (**d**) After miR-129-5p inhibitor was transfected into DPCs, the effect of it on the proliferation of DPCs was determined after 0, 24, 48, and 72 h. ** *p* < 0.01. DPCs: dermal papilla cells; NC: negative control. In (**a**, **b**), the first quadrant represents the cell debris, the second quadrant represents late apoptotic cells, the third quadrant is living cells, and the fourth quadrant is early apoptotic cells. The apoptotic rate is Q2 + Q4.

**Figure 5 genes-13-00679-f005:**
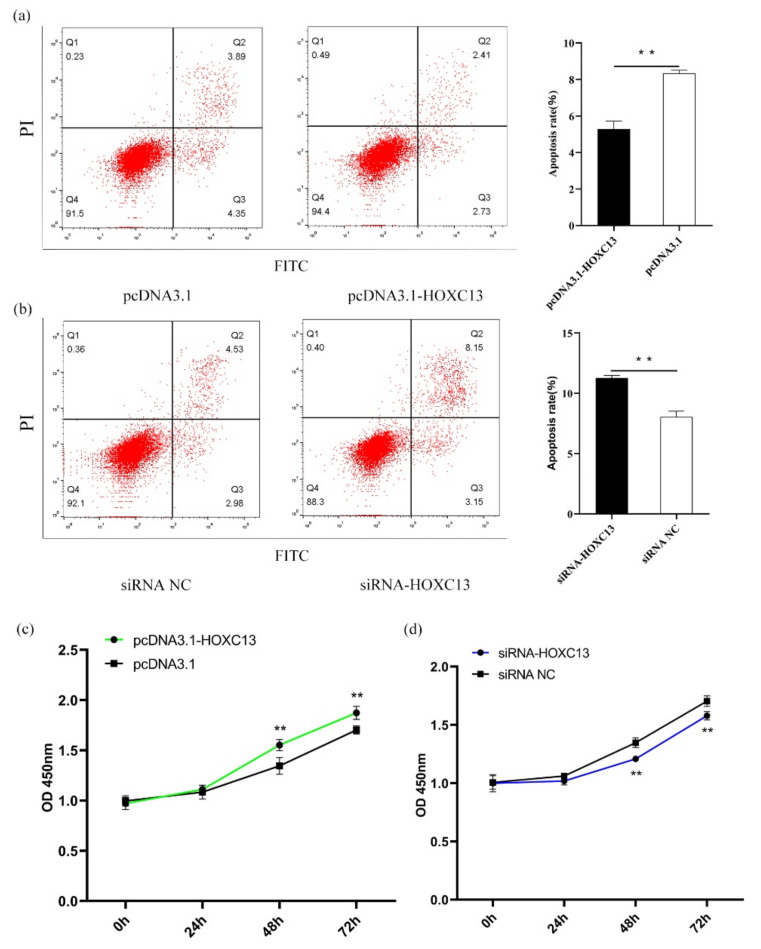
*HOXC13* inhibited apoptosis and promoted the proliferation of DPCs. (**a**) *HOXC13* inhibited apoptosis of DPCs. (**b**) SiRNA-HOXC13 promoted apoptosis of DPCs. (**c**) After pcDNA3.1-HOXC13 was transfected into DPCs, the effect of *HOXC13* on the proliferation of DPCs was determined after 0, 24, 48, and 72 h. (**d**) After siRNA-HOXC13 was transfected into DPCs, the effect of HOXC13 on the proliferation of DPCs was determined after 0, 24, 48, and 72 h. ** *p* < 0.01. DPCs: dermal papilla cells; NC: negative control.

**Table 1 genes-13-00679-t001:** Primer sequences used in qPCR.

Genes	Primers Sequence(5′→3′)
*GAPDH*	F: TCACCATCTTCCAGGAGCGA R: CACAATGCCGAAGTGGTCGT
*BCL2*	F: ACATCGCCCTGTGGATGACTG R: CGAGGGTGATGCAAGCTCCTAT
*WNT2*	F: AGCCATCCAGGTCGTCATGAACCAG R: TGCACACACGACCTGCTGTACCC
*CCND1*	F: GAACGCTACCTTCCCCAGTGCTC R: CCTCACAGACCTCCAGCATCCAG
*SFRP2*	F: CCAGCCCGACTTCTCCTACAAGC R: TCCAGCACCTCTTTCATGGTCT
*TGF-β1*	F: CAGGTCCTTGCGGAAGTCAA R: CTGGAACGGGCTCAACATCTA
*FGF2*	F: GTGTGTGCAAACCGTTACCTT R: TCGTTTCAGTGCCACATACCAG
*LEF1*	F: CATCTCGGGTGGATTCAGG R: ATGAGGGATGCCAGTTGTG
*HOXC13*	F: TGGAGAAAGAGTACGCAGCC R: TCTTCTCTTTGACTCGGCGG
*miR-129-5p*	CTTTTTGCGGTCTGGGCTTGC
*U6*	CAAGGATGACACGCAAATTCG

Note: F: forward primer; R: reverse primer.

## Data Availability

Forecast of targeted binding sites between miR-129-5p and *HOXC13*. Available online: https://bibiserv.cebitec.uni-bielefeld.de/ (accessed on 11 April 2022).

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
