# Peer review of "miR-129-5p Participates in Hair Follicle Growth by Targeting HOXC13 in Rabbit"

_genes, 2022, doi:10.3390/genes13040679_

Round 1

Reviewer 1 Report

The paper has been improved

Author Response

Dear Academic Editor,

Thank you again for your letter and for the academic editor’s comments concerning our manuscript entitled “miR-129-5p Participates Hair Follicles Growth by Targeting HOXC13 in Rabbit” (ID: Genes-1669057). Those comments are all valuable and very helpful for revising and improving our paper. We have studied comments carefully and have made correction which we hope meet with approval. The main corrections in the paper and the responds to the academic editor’s comments are as following:

I used the “Track Changes” function to modify the tense and grammar of the article.

Thanks for all the help.

Best wishes,

Fan Yao

Apr 6, 2022

Reviewer 2 Report

Authors should prepare a minor revision for publish in second review.

  1. In figure1 b, why are there differences between miRNA mimics NC and inhibitor NC? It should be same with  figure2 f.
  2. Please carefully check writing! ex. Line 177.
  3. Do authors try to identify the function of HFDRGs? Perhaps to try to make GO analysis?
  4. Please make sure miR-129-5p primers! Please add more details if authors use kit to detect its expression.

Author Response

This manuscript is a resubmission of an earlier submission. The following is a list of the peer review reports and author responses from that submission.

Round 1

Reviewer 1 Report

The structure of some sentences is wrong and it is hard to understand what the authors want to express. Use of the indefinite article "the" should be revised and corrected throughout the text.

The abstract should be rewritten. 
The phrasing in the introduction is very poor; a proper rationale for the study is hard to get from the text. 
Methods:
What do you mean by this in the methods section: "This study selected 15 healthy rabbits of the same weight. To promote cell death, the rabbits were administered with Zoletil-50 (6mg/kg) injection into the ear vein, ensuring that the muscle relaxants used in anesthesia did not inhibit respiration" ?
Do you mean killing of rabbits (euthanasia)? were the rabbits killed by overdose or other methods were used to provoke death?
The description of the methods is very incomplete. From reading, one cannot know how DC cells were exactly isolated and how their identity was secured (DC vs fibroblasts from skin) 
How were DC cells generated and how was RNA extracted from these cells and from rabbit tissues for qPCR?
RAB-9 cells are fibroblasts and not DPCs!

Results:

Bioinformatic analysis:

what supports this statement:  It further suggested miR-129-5p to possibly have similar biological functions?

Why did you select HOXC13 for further analysis out of  244 common target genes?

Figure 1: the resolution is very poor.

Analysis of the expression pattern of miR-129-5p and Figure 2:

Results from Figure 2 are not related to each other. A shows expression of miR-129-5p in different tissues. Since the paper states that miR-129-5p influences DP biology, is it expressed in DP? here the authors should include evidence of in situ expression either with ISH or isolating DPCs (for example FACS sorting) and then measuring expression with PCR.

B to E are not related to A, should therefore be on a different figure

B then shows effect of transfection of miR-129-5p in DPCs regarding expression of different genes related to apoptosis and HF biology, C shows miR-129-5p expression in DPCs after transfection, D shows effect of inhibition of miR-129-5p on genes studied in B. Are these experiments done with primary DPCs or with RAB-9 cells?
E shows, I assume, expression (qPCR) of HOXC13 in DPCs transfected with anti-miR or control anti-miR. This result is not described in the text. In addition, this result shows an induction of HOXC13 expression upon miR-129-5p inhibition compared to control, which should not occur since anti-miR treatment does not interfere with transcription. How can you explain this?

The miR-129-5p enhances apoptosis and inhibits the proliferation of the DPCs and Figure3:

Very poorly written. Resolution of figure is very poor.

Texas red and FITC are marking what? Dead/DNA and Annexin?

Growth of DPCs should be replaced in text with proliferation

The miR-129-5p regulates the follicle development of the rabbit hair by targeting HOXC13 and Figure 4:

The setence "To determine that miR-129-5p targeting the HOXC13, the HOXC13 3′-UTR sequence was obtained from NCBI" has to be rewritten. 

Are all experiments fr this figure done with primary DPCs or with RAB-9 cells? 

b and c show the same, redundant information! 

The results reported here do not support the statement "miR-129-5p regulates the follicle development of the rabbit hair by targeting HOXC13". It only shows that miR-129-5p targets HOXC13. Whether this is a mechanism present in HF was not investigated.

Also, the effect on gene expression shown in figure 2 are due to HOXC13 being targeted by miR-129-5p or this is the result of mir-129-5p targeting any of the other 244 predicted genes?

Discussion: poorly written. Some conclusions are not substantiated by data.

Overall, the paper should be renamed to express the findings of miR-129-5p targeting HOXC13, whether this affects DPCs and HF biology can only be speculated with the data presented here.

The amount of presented data is very scarce to justify publication as an original article. The authors should consider other form of publication such as short communication. 

Author Response

Dear Reviewer,

Thank you again for your comments concerning our manuscript entitled “miR-129-5p Inhibits the Proliferation of the Dermal Papilla Cells by Targeting HOXC13 in Rabbit” (ID: Genes-1561936). Those comments are all valuable and very helpful for revising and improving our paper. We have studied comments carefully and have made correction which we hope meet with approval. Revised portion are marked in red in the paper. The main corrections in the paper and the responds to the academic editor’s comments are as following:

I made a lot of modifications to the paper.

  1. The summary section was rewritten.
  2. The summary of the second half of the introduction has been modified to make it more organized.
  3. Some experimental methods were added.
  4. A large number of modifications were made to the results, and the original figure 2a was deleted. The expressions of miR-129-5p and HOXC13 in different hair follicle development cycles of Angora rabbits were replaced and the correlation analysis was carried out. Increased HOXC13 protein result, and the effect of HOXC13 on the expression of related genes to DPCs. There results were support the conclusions of this paper.
  5. The discussion part is modified to make it more organized.
  6. Two references were added.

Specific comments:

  1. The structure of some sentences is wrong and it is hard to understand what the authors want to express. Use of the indefinite article "the" should be revised and corrected throughout the text.

Response: Many thanks to the reviewers for their comments. According to your comments, I asked experts to edit the full text grammar, punctuation, spelling and overall style.

  1. The abstract should be rewritten. The phrasing in the introduction is very poor; a proper rationale for the study is hard to get from the text. 

Response: Many thanks to the reviewers for their comments. According to your comments, I have rewritten the abstract and revised the introduction to make it more structured.

  1. What do you mean by this in the methods section: "This study selected 15 healthy rabbits of the same weight. To promote cell death, the rabbits were administered with Zoletil-50 (6mg/kg) injection into the ear vein, ensuring that the muscle relaxants used in anesthesia did not inhibit respiration"?
    Do you mean killing of rabbits (euthanasia)? were the rabbits killed by overdose or other methods were used to provoke death?
    The description of the methods is very incomplete. From reading, one cannot know how DC cells were exactly isolated and how their identity was secured (DC vs fibroblasts from skin) 
    How were DC cells generated and how was RNA extracted from these cells and from rabbit tissues for qPCR?
    RAB-9 cells are fibroblasts and not DPCs!

Response: Many thanks to the reviewers for their comments. According to your comments, I have added relevant experimental methods in the article (Please see 2.2 Cell culture and transfection assay). The rabbits were administered with Zoletil-50 (6mg/kg) injection into the ear vein, ensuring that the muscle relaxants used in anesthesia did not inhibit respiration, then the tissue samples of rabbit back skin on days 30, 60, 90, 120, 130, and 150 were collected. The rabbits were killed after the last sampling. At the same time, all the cell experiments in this paper were used cell lines, the DPCs were the cell lines successfully established in our laboratory [1].

[1] Li, J.L.; Zhao, B.H.; Zhang, C.; Zhang, X.Y.; Dai, Y.Y.; Hu, S.S.; Yang, N.S.; Chen, Y.; Wu, X.S. Establishment and Functional Characterization of Immortalized Rabbit Dermal Papilla Cell Lines. 2021.

  1. Bioinformatic analysis: what supports this statement:  It further suggested miR-129-5p to possibly have similar biological functions?

Why did you select HOXC13 for further analysis out of 244 common target genes?

Figure 1: the resolution is very poor.

Results from Figure 2 are not related to each other. A shows expression of miR-129-5p in different tissues. Since the paper states that miR-129-5p influences DP biology, is it expressed in DP? here the authors should include evidence of in situ expression either with ISH or isolating DPCs (for example FACS sorting) and then measuring expression with PCR.

Response: Many thanks to the reviewers for their comments. According to your comments, I reorganized the results. The new first chapter was named “Prediction of targeting relationship between miR-129-5p and HOXC13”, at the same time, thanks for your reminder, I reanalyzed the results and found that this statement “It further suggested miR-129-5p to possibly have similar biological functions” was wrong. The results in figure 1a only showed that miR-129-5p was conserved among various organisms, so I deleted the wrong conclusion. In addition, I recreated Figure 1 to improve its clarity.

  1. Analysis of the expression pattern of miR-129-5p and Figure 2:

Results from Figure 2 are not related to each other. A shows expression of miR-129-5p in different tissues. Since the paper states that miR-129-5p influences DP biology, is it expressed in DP? here the authors should include evidence of in situ expression either with ISH or isolating DPCs (for example FACS sorting) and then measuring expression with PCR.

B to E are not related to A, should therefore be on a different figure

B then shows effect of transfection of miR-129-5p in DPCs regarding expression of different genes related to apoptosis and HF biology, C shows miR-129-5p expression in DPCs after transfection, D shows effect of inhibition of miR-129-5p on genes studied in B. Are these experiments done with primary DPCs or with RAB-9 cells?

E shows, I assume, expression (qPCR) of HOXC13 in DPCs transfected with anti-miR or control anti-miR. This result is not described in the text. In addition, this result shows an induction of HOXC13 expression upon miR-129-5p inhibition compared to control, which should not occur since anti-miR treatment does not interfere with transcription. How can you explain this?

Response: Many thanks to the reviewers for their comments. According to your comments, the original Figure 2a is not closely related to the article, so I deleted it. In addition, I changed “3.2 Analysis of the expression pattern of miR-129-5p” to “3.3miR-129-5p targeted the HOXC13 to regulate the expression of the HF development-related genes”, and increased the expression of miR-129-5p and HOXC13 in different hair follicle development stages (Please see figure 3a) and the effect of HOXC13 on the expression of hair follicle development related genes in DPCs (Please see figure 3b and 3c). The results showed that miR-129-5p was highly expressed in catagen and negatively correlated with HOXC13 (Pearson’s R =-0.313, P < 0.05). Due to the Chinese lunar spring vacation, I may not have time to do these experiments for the supplement evidence of situ expression either with ISH or isolating DPCs (for example FACS sorting) and then measuring expression with PCR that you mentioned, so I am very sorry for not providing the results of these experiments. Furthermore, these experiments were performed using DPCs, while RAB-9 cells were only used for dual-luciferase report assay. Otherwise, the original Figure 2e is the error I added when drawing, which has been corrected. In addition, miRNA can also inhibit the expression of target genes by reducing the stability of mRNA [2], such as miR-590-5p, which reduces the expression level of LOX-1 [3].

[2] Bartel D P. MicroRNAs: target recognition and regulatory functions[J]. cell, 2009, 136(2): 215-233.

[3] Qin B, Xiao B, Jiang T, Yang H. [Effects of miR-590-5p on ox-LDL-induced endothelial cells apoptosis and LOX-1 expression]. Zhong nan da xue xue bao. Yi xue ban = Journal of Central South University. Medical Sciences. 2012 Jul;37(7):675-681. DOI: 10.3969/j.issn.1672-7347.2012.07.005. PMID: 22886218.

  1. The miR-129-5p enhances apoptosis and inhibits the proliferation of the DPCs and Figure3:

Very poorly written. Resolution of figure is very poor

Texas red and FITC are marking what? Dead/DNA and Annexin?

Growth of DPCs should be replaced in text with proliferation

 Response: Many thanks to the reviewers for their comments. According to your comments, I modified this result and redesigned it to improve resolution, and I changed “3.3 The miR-129-5p enhances apoptosis and inhibits the proliferation of the DPCs” to “3.4 miR-129-5p enhances apoptosis and inhibits the proliferation of the DPCs”. FITC labeled Annexin V, which can identify early apoptotic cells. Propidium Iodide (PI) is a nucleic acid dye. PI cannot penetrate the complete cell membrane of normal cells or early apoptotic cells, but it can penetrate the cell membrane of cells damaged by membrane, such as late apoptotic cells or necrotic cells, and bind to the DNA in them, so that the nucleus is stained red, which can be used to distinguish early apoptotic cells from late apoptotic cells or necrotic cells. In figure 4a and 4b, the first quadrant represents the cell debris, the second quadrant represents late apoptotic cells, the third quadrant is living cells, and the fourth quadrant is early apoptotic cells. The apoptotic rate is Q2 + Q4. Furthermore, I have replaced growth of DPC with proliferation of DPC.

  1. The miR-129-5p regulates the follicle development of the rabbit hair by targeting HOXC13 and Figure 4:

The setence "To determine that miR-129-5p targeting the HOXC13, the HOXC13 3′-UTR sequence was obtained from NCBI" has to be rewritten.

Are all experiments fr this figure done with primary DPCs or with RAB-9 cells?

b and c show the same, redundant information! 

Response: Many thanks to the reviewers for their comments. According to your comments, I modified the language description and added the Western Blot result (Please see figure 2e), and I changed “3.4 The miR-129-5p regulates the follicle development of the rabbit hair by targeting HOXC13 and Figure 4” to “3.2 Verification of the targeting relationship between miR-129-5p and HOXC13”. In addition, RAB-9 cells were only used for dual-luciferase reporter assay, and DPC cells were used in other experiments.

  1. The results reported here do not support the statement "miR-129-5p regulates the follicle development of the rabbit hair by targeting HOXC13". It only shows that miR-129-5p targets HOXC13. Whether this is a mechanism present in HF was not investigated.

Also, the effect on gene expression shown in figure 2 are due to HOXC13 being targeted by miR-129-5p or this is the result of mir-129-5p targeting any of the other 244 predicted genes?

Response: Many thanks to the reviewers for their comments. According to your comments, according to your suggestion, we detected and analyzed the expression of HOXC13 and miR-129-5p in the skin of Angora rabbits at different hair follicle development stages. The results showed that miR-129-5p was highly expressed in catagen, and HOXC13 was highly expressed in anagen, which was negatively correlated with HOXC13 (Pearson’s R =-0.313, P < 0.05) (Figure 3a). In addition, Western Blot results also showed that miR-129 - 5p could regulate the expression of HOXC13 protein (Figure 2e). These results showed that miR-129 - 5p could target HOXC13 and regulate the periodic development of hair follicles by targeting HOXC13. Furthermore, DPCs are indispensable cells in the periodic development of HF, which provide the necessary nutrition and signal support for the growth of HF. Therefore, the effects of miR-129 – 5p and HOXC13 on the expression of genes related to HF development in DPCs, and the effects of miR-129-5p on the proliferation and apoptosis of DPCs were verified by qRT-PCR, indicating that miR-129-5p might regulate the expression of the related genes by targeting HOXC13, thereby affecting the proliferation and apoptosis of the DPCs, and then regulating the periodic development of HF.

  1. Discussion: poorly written. Some conclusions are not substantiated by data.

Response: Many thanks to the reviewers for their comments. According to your comments, I rewrite the conclusion part to make it more complete. At the same time, I add some experiments to make up for the deficiency of the results, hoping that these results I add can make my conclusion more complete.

Thanks for all the help, your comments will enrich my article.

Best wishes,

Fan Yao

Feb 3, 2022

Reviewer 2 Report

Review for Genes Yao et al.,

This is an interesting paper looking at the role of miR129 in hair follicle development. The story fits into the literature. The authors cite relevant articles from Elaine Fuch’s laboratory and others that have been involved in deciphering how the hair follicle forms. Before this paper can be accepted for publication, however, there are a few comments that need to be addressed.

Major

  • Please define the concentrations of mimics and inhibitors you used. I looked in the materials and methods as well as the figure legends and results section but I could not find these data. That said, once you add the concentration numbers, please add a qualifying sentence as to why you used that concentration. Can you calculate the concentration of miR129 in Figure 2a skin and use that for transfection? See (b) below.
  • I believe the results need to be done with two or three different dosages (concentrations) if mimics and inhibitors. Although the results show significance, the changes in gene expression are subtle at the concentration being used. Why would that subtle difference in gene expression alter or effect hair follicle formation? Again, if you used the same concentration that is in skin in figure 2a, what would that look like with respect to gene expression? Please don’t sacrifice any more rabbits for this. I hope you have a cell line or some frozen cells you can do these experiments with.
  • Figure 3…please give more detail about what the figure is showing especially in A, What is each quadrant referring to? I know it says apoptosis, but what in the figure tells me that? The materials and methods using the apoptosis detection kit does not tell me much. Which quadrant do you want me to assess? I am assuming it’s the upper right, but what’s happening in the upper left or lower right? Are those quadrants even necessary?

Minor

  • The manuscript is written pretty well, however, you use The’s in the wrong place sometimes. First sentence of abstract has too many The’s. In the introduction line 50 page 2 The HoxC13… no the is needed. There are more like that throughout the text.
  • Pgae 2 line 53 Should that be negatively affecting? Confusing sentence
  • Page 2 line 55…at the same time, it inhibits…What is ‘it’?
  • Figure 1 please define the colors within the sequences. Wat does red mean? Purple? Blue?
  • Figure 2 How many repeats were done to get the results in 2a? Needs and N value
  • Please state if the DPCs were passaged at all or if they were straight from the rabbit and used once they were 80% confluent. It’s OK if you passaged a few times.

Author Response

Thank you again for your comments concerning our manuscript entitled “miR-129-5p Inhibits the Proliferation of the Dermal Papilla Cells by Targeting HOXC13 in Rabbit” (ID: Genes-1561936). Those comments are all valuable and very helpful for revising and improving our paper. We have studied comments carefully and have made corrections which we hope meet with approval. Revised portions are marked in red in the paper. The main corrections in the paper and the responds to the academic editor’s comments are as following:

I made a lot of modifications to the paper.

  1. The summary section was rewritten.
  2. The summary of the second half of the introduction has been modified to make it more organized.
  3. Some experimental methods were added.
  4. A large number of modifications were made to the results, and the original figure 2a was deleted. The expressions of miR-129-5p and HOXC13 in different hair follicle development cycles of Angora rabbits were replaced and the correlation analysis was carried out. Increased HOXC13 protein result, and the effect of HOXC13 on the expression of related genes to DPCs. There results were support the conclusions of this paper.
  5. The discussion part is modified to make it more organized.
  6. Two references were added.

Specific comments:

  1. Please define the concentrations of mimics and inhibitors you used. I looked in the materials and methods as well as the figure legends and results section but I could not find these data. That said, once you add the concentration numbers, please add a qualifying sentence as to why you used that concentration. Can you calculate the concentration of miR129 in Figure 2a skin and use that for transfection? See (b) below.

Response: Many thanks to the reviewers for their comments. According to your comments, I have added relevant concentrations in the article (see 2.2 Cell culture and transfection assay). In addition, figure 2a is not relevant to this article, so I replaced the expression of miR-129-5p and HOXC13 in different hair follicle development stage as shown in Figure 3a

  1. I believe the results need to be done with two or three different dosages (concentrations) if mimics and inhibitors. Although the results show significance, the changes in gene expression are subtle at the concentration being used. Why would that subtle difference in gene expression alter or effect hair follicle formation? Again, if you used the same concentration that is in skin in figure 2a, what would that look like with respect to gene expression? Please don’t sacrifice any more rabbits for this. I hope you have a cell line or some frozen cells you can do these experiments with.

Response: Many thanks to the reviewers for their comments. According to the instructions, I used three different concentrations in the pre-experiment. Among them, the promotion or inhibition of miR-129-5p was the best after transfection according to the concentration used in the article. These genes are hair follicle development related genes, maybe individual gene changes have little effect on hair follicles, but a large number of gene changes may change the formation of hair follicles. At the same time, all the cell experiments in this paper were used cell lines, the DPCs were the cell lines successfully established in our laboratory.

Li, J.L.; Zhao, B.H.; Zhang, C.; Zhang, X.Y.; Dai, Y.Y.; Hu, S.S.; Yang, N.S.; Chen, Y.; Wu, X.S. Establishment and Functional Characterization of Immortalized Rabbit Dermal Papilla Cell Lines. 2021.

3. Figure 3…please give more detail about what the figure is showing especially in A, what is each quadrant referring to? I know it says apoptosis, but what in the figure tells me that? The materials and methods using the apoptosis detection kit does not tell me much. Which quadrant do you want me to assess? I am assuming it’s the upper right, but what’s happening in the upper left or lower right? Are those quadrants even necessary?

Response: Many thanks to the reviewers for their comments. In figure 4a and 4b, the first quadrant represents the cell debris, the second quadrant represents late apoptotic cells, the third quadrant is living cells, and the fourth quadrant is early apoptotic cells. The apoptotic rate is Q2 + Q4.

4. The manuscript is written pretty well, however, you use The’s in the wrong place sometimes. First sentence of abstract has too many The’s. In the introduction line 50 page 2 The HoxC13… no the is needed. There are more like that throughout the text.

Pgae 2 line 53 Should that be negatively affecting? Confusing sentence

Response: Many thanks to the reviewers for their comments. According to your comments, I have rewritten the abstract and other parts, and carried out the polishing to make it more in line with the usage habits of English-speaking countries.

5. Figure 1 please define the colors within the sequences. Wat does red mean? Purple? Blue? Figure 2 How many repeats were done to get the results in 2a? Needs and N value

Please state if the DPCs were passaged at all or if they were straight from the rabbit and used once they were 80% confluent. It’s OK if you passaged a few times.

Response: Many thanks to the reviewers for their comments. According to your comments, I have annotated the meaning of different colors in Figure 1, blue is base C, red is base U, purple is base G. The original Figure 2a is not closely related to the article, so I deleted it. The DPCs were the cell lines successfully established in our laboratory.

Thanks for all the help.

Best wishes,

Fan Yao

Feb 3, 2022

Reviewer 3 Report

While the authors provide some level of evidence that miR-129-5p is capable of interacting with the Hoxc13 via a 7 nucleotide region of homology in its UTR, the data do not in any way conclusively support the claim that this is of in vivo relevance. The data were exclusively obtained by using in vitro assays with RAB-9 fibroblasts or rabbit dermal papilla cells (DPCs). As an intact DP is known to be in constant and dynamic cross-talk with surrounding ectodermal cells during hair cycle control it is difficult to see how data derived from experiments with dissociated DPCs can significantly contribute to understanding miR-129-5p function in an in vivo DP. Furthemore, as there is little if any evidence for Hoxc13 mRNA expression in the DP (which would be a precondition for this presumptive Hoxc13/miR-129-5p regulatory relationship) it would be helpful if the authors could provide a valid reference.      

Author Response

Dear Reviewer,

Thank you again for your comments concerning our manuscript entitled “miR-129-5p Inhibits the Proliferation of the Dermal Papilla Cells by Targeting HOXC13 in Rabbit” (ID: Genes-1561936). Those comments are all valuable and very helpful for revising and improving our paper. We have studied comments carefully and have made correction which we hope meet with approval. Revised portion are marked in red in the paper. The main corrections in the paper and the responds to the academic editor’s comments are as following:

I made a lot of modifications to the paper.

  1. The summary section was rewritten.
  2. The summary of the second half of the introduction has been modified to make it more organized.
  3. Some experimental methods were added.
  4. A large number of modifications were made to the results, and the original figure 2a was deleted. The expressions of miR-129-5p and HOXC13 in different hair follicle development cycles of Angora rabbits were replaced and the correlation analysis was carried out. Increased HOXC13 protein result, and the effect of HOXC13 on the expression of related genes to DPCs. There results were support the conclusions of this paper.
  5. The discussion part is modified to make it more organized.
  6. Two references were added.

Specific comments:

  1. While the authors provide some level of evidence that miR-129-5p is capable of interacting with the Hoxc13 via a 7 nucleotide region of homology in its UTR, the data do not in any way conclusively support the claim that this is of in vivo relevance. The data were exclusively obtained by using in vitro assays with RAB-9 fibroblasts or rabbit dermal papilla cells (DPCs). As an intact DP is known to be in constant and dynamic cross-talk with surrounding ectodermal cells during hair cycle control it is difficult to see how data derived from experiments with dissociated DPCs can significantly contribute to understanding miR-129-5p function in an in vivo DP. Furthermore, as there is little if any evidence for Hoxc13 mRNA expression in the DP (which would be a precondition for this presumptive Hoxc13/miR-129-5p regulatory relationship) it would be helpful if the authors could provide a valid reference.

Response: Many thanks to the reviewers for their comments. According to your suggestion, we detected and analyzed the expression of HOXC13 and miR-129-5p in the skin of Angora rabbits at different hair follicle development stages. The results showed that miR-129-5p was highly expressed in catagen, and HOXC13 was highly expressed in anagen, which was negatively correlated with HOXC13 (Pearson’s R =-0.313, P < 0.05) (Figure 3a). In addition, Western Blot results also showed that miR-129 - 5p could regulate the expression of HOXC13 protein (Figure 2e). These results showed that miR-129 - 5p could target HOXC13 and regulate the periodic development of hair follicles by targeting HOXC13. Furthermore, DPCs are indispensable cells in the periodic development of HF, which provide the necessary nutrition and signal support for the growth of HF. Therefore, the effects of miR-129 – 5p and HOXC13 on the expression of genes related to HF development in DPCs, and the effects of miR-129-5p on the proliferation and apoptosis of DPCs were verified by qRT-PCR, indicating that miR-129-5p might regulate the expression of the related genes by targeting HOXC13, thereby affecting the proliferation and apoptosis of the DPCs, and then regulating the periodic development of HF.

Thanks for all the help.

Best wishes,

Fan Yao

Feb 3, 2022

Round 2

Reviewer 1 Report

The manuscript has been improved, however there are still some areas regarding redaction and semantics which must be improved. 

The abtract section requires further editing. Its structure does not provide a clear rationale for the study. The style of writing should be also reviewed.

Please include figure 4 a and b labels for Annexin V FITC and PI Texas red

In the first review I wrote: Since the paper states that miR-129-5p influences DP biology, is it expressed in DP? here the authors should include evidence of in situ expression either with ISH or isolating DPCs (for example FACS sorting) and then measuring expression with PCR.

Your answer is: Due to the Chinese lunar spring vacation, I may not have time to do these experiments for the supplement evidence of situ expression either with ISH or isolating DPCs (for example FACS sorting) and then measuring expression with PCR that you mentioned, so I am very sorry for not providing the results of these experiments.

If you have stablished DPCs lines it should be easy to generate material for PCR so at least there is some proof of expression of miR-129-5p in DPCs. 

What do you mean by this in the discussion: "Our research has revealed miR-129-5p is to be conserved among various species". Where is the data supporting this statement?

Please allow for someone to read the manuscript before resubmission to check for soundness of writing. 

Author Response

Dear Academic Editor,

Thank you again for your letter and for the academic editor’s comments concerning our manuscript entitled “miR-129-5p Inhibits the Proliferation of the Dermal Papilla Cells by Targeting HOXC13 in Rabbit” (ID: Genes-1561936). Those comments are all valuable and very helpful for revising and improving our paper. We have studied comments carefully and have made correction which we hope meet with approval. The main corrections in the paper and the responds to the academic editor’s comments are as following:

Specific comments:

  1. The manuscript has been improved, however there are still some areas regarding redaction and semantics which must be improved.

Response: Many thanks to the reviewers for their comments. According to your opinion, I modified the tense and grammar of the article, and invited experts to review the grammar of the article.

  1. The abstract section requires further editing. Its structure does not provide a clear rationale for the study. The style of writing should be also reviewed.

Response: Many thanks to the reviewers for their comments. According to your opinion, I have rewritten the abstract.

  1. Please include figure 4 a and b labels for Annexin V FITC and PI Texas red

Response: Many thanks to the reviewers for their comments. According to your opinion, I remade the drawing and add the abscissa FITC and ordinate PI to figure 4a and 4b.

  1. If you have stablished DPCs lines it should be easy to generate material for PCR so at least there is some proof of expression of miR-129-5p in DPCs.

Response: Many thanks to the reviewers for their comments. In the early stage, we have established rabbit hair dermal papilla cell lines and used the FACS sorting method for identification. The relevant pictures are attached for reference (Please see figure A and B). Since no ISH or FACS sorting experiment was designed at that time to detect the expression levels of skin samples at different stages in different periods of rabbit skin, only qRT-PCR was used to detect the expression levels in the skin, resulting in the use of skin samples with periodic hair follicle development in rabbit. So, I am prepared to detect the expression level of miR-129 - 5p in melanocytes, RAB-9 cells, and DPCs to prove that the expression level of miR-129 - 5p in DPCs is significantly higher than that in other cells. However, only DPCs have been recovered in these days, and other cells have not recovered so far. Therefore, these experiments have not been carried out. Therefore, we transfected HOXC13 and siRNA-HOXC13 into dermal papilla cells and detected the proliferation and apoptosis of dermal papilla cells. The results showed that HOXC13 could promote the proliferation of dermal papilla cells and inhibit the apoptosis of dermal papilla cells, which was contrary to the results of miR-129 - 5p, indicating that miR-129 - 5p could promote the apoptosis of dermal papilla cells and inhibit their proliferation by targeting HOXC13. The results are shown figure C below.

Figure A: Morphology of the seven DPC lines and primary DPCs. (a) Isolation and morphological observations of primary DPCs (Scale Bar: 100 μm). (b) Fluorescence analysis of pLVX-IRES-Puro-GFP transfection after 24h, 48h. (Scale Bar: 100 μm). (c) Cell morphologies (Scale Bar: 100 μm). (d) Alkaline phosphatase analysis. Primary DPCs and immortalized cell lines were used for ALPL detection (Scale Bar: 50 μm). Pri-DPC: primary dermal papilla cell

Figure B: Indirect immunofluorescence identification of dermal papilla cells

Figure C. HOXC13 inhibits apoptosis and promotes the proliferation of DPCs. (a) HOXC13 inhibits apoptosis of DPC. (b) siRNA-HOXC13 promotes apoptosis of DPC. (c) After pcDNA3.1-HOXC13 were transfected into DPCs, the effect of HOXC13 on DPC proliferation was determined after 0, 24, 48, and 72 h. (d) After siRNA-HOXC13 was transfected into DPCs, the effect of HOXC13 on DPC proliferation was determined after 0, 24, 48, and 72 h. **p < 0.01. DPC: dermal papilla cell; NC: negative control. In (a) and (b), the first quadrant represents the cell debris, the second quadrant represents late apoptotic cells, the third quadrant is living cells, and the fourth quadrant is early apoptotic cells. The apoptotic rate is Q2 + Q4.

  1. What do you mean by this in the discussion: "Our research has revealed miR-129-5p is to be conserved among various species". Where is the data supporting this statement?

Response: Many thanks to the reviewers for their comments. I ' m sorry, I missed a word behind this sentence. The complete sentence should be “In current study revealed that miR-129-5p is to be conserved among various species by Using MEGA7.0 software”. Figure 1a supports this conclusion.

  1. Please allow for someone to read the manuscript before resubmission to check for soundness of writing.

 Response: Many thanks to the reviewers for their comments. According to your comments, I modified the tense and grammar of the article, and invited experts to review the grammar of the article.

Thanks for all the help, your comments will enrich my article.

Best wishes,

Fan Yao

Feb 19, 2022

Reviewer 2 Report

The authors have addressed all of my comments appropriately.

Reviewer 3 Report

My main concern was the lack of data showing miR-129-5p and Hoxc13 expression in native dermal papillae at different stages of the hair cycle, respectively, as implied by the way how the paper was written. By showing miR-129-5p and Hoxc13 in skin samples and hair follicles this concern has partially been addressed but also somewhat "side-stepped" - this needs to be further elaborated. 

The data showing an inverse relationship between miR-129-5p and Hoxc13 expression and the molecular interaction between the two is novel and interesting. But it needs to be elaborated beyond the nearly exclusive focus on dermal papilla cells (DPCs). Some in situ or immunolabeling data should be included. A summary cartoon showing miR-129-5p-Hoxc13 interaction in the keratinocytes of the hair follicle matrix surrounding the DP (I suspect this is where this interaction takes place) and how this may signal back to the DP, etc. would be extremely helpful. I believe the authors should be encouraged to develop a new manuscript along these lines which would also require changing the title (as currently written it's misleading). Aside from the this, the authors should be encouraged to have the paper reviewed by a native English speaker before submission - the syntax is frequently off and unacceptable, already in the Abstract. 

Bottom line, the manuscript continues to be a construction site and is far from ready for publication. But the authors should be encouraged to develop and present a conclusive case for a NEW submission by building on the interesting findings they have in hand.

Author Response

Dear Academic Editor,

Thank you again for your letter and for the academic editor’s comments concerning our manuscript entitled “miR-129-5p Inhibits the Proliferation of the Dermal Papilla Cells by Targeting HOXC13 in Rabbit” (ID: Genes-1561936). Those comments are all valuable and very helpful for revising and improving our paper. We have studied comments carefully and have made correction which we hope meet with approval. The main corrections in the paper and the responds to the academic editor’s comments are as following:

I made a lot of modifications to the paper.

  1. The summary section was rewritten.
  2. we transfected HOXC13 and siRNA-HOXC13 into dermal papilla cells and detected the proliferation and apoptosis of dermal papilla cells. The results showed that HOXC13 could promote the proliferation of dermal papilla cells and inhibit the apoptosis of dermal papilla cells, which was contrary to the results of miR-129 - 5p, indicating that miR-129 - 5p could promote the apoptosis of dermal papilla cells and inhibit their proliferation by targeting HOXC13. The results are shown figure below.

Figure: HOXC13 inhibits apoptosis and promotes the proliferation of DPCs. (a) HOXC13 inhibits apoptosis of DPC. (b) siRNA-HOXC13 promotes apoptosis of DPC. (c) After pcDNA3.1-HOXC13 were transfected into DPCs, the effect of HOXC13 on DPC proliferation was determined after 0, 24, 48, and 72 h. (d) After siRNA-HOXC13 was transfected into DPCs, the effect of HOXC13 on DPC proliferation was determined after 0, 24, 48, and 72 h. **p < 0.01. DPC: dermal papilla cell; NC: negative control. In (a) and (b), the first quadrant represents the cell debris, the second quadrant represents late apoptotic cells, the third quadrant is living cells, and the fourth quadrant is early apoptotic cells. The apoptotic rate is Q2 + Q4.

Thanks for all the help.

Best wishes,

Fan Yao

Feb 19, 2022
